# Semi-Supervised Audio Representation Learning for Modeling Beehive Strengths

## Abstract

Honey bees are critical to our ecosystem and food security as a pollinator, contributing 35% of our global agriculture yield (Klein et al., 2007). In spite of their importance, beekeeping is exclusively dependent on human labor and experience-derived heuristics, while requiring frequent human checkups to ensure the colony is healthy, which can disrupt the colony. Increasingly, pollinator populations are declining due to threats from climate change, pests, environmental toxicity, making their management even more critical than ever before in order to ensure sustained global food security. To start addressing this pressing challenge, we developed an integrated hardware sensing system for beehive monitoring through audio and environment measurements, and a hierarchical semi-supervised deep learning model, composed of an audio modeling module and a predictor, to model the strength of beehives. The model is trained jointly on audio reconstruction and prediction losses based on human inspections, in order to model both low-level audio features and circadian temporal dynamics. We show that this model performs well despite limited labels, and can learn an audio embedding that is useful for characterizing different sound profiles of beehives. This is the first instance to our knowledge of applying audio-based deep learning to model beehives and population size in an observational setting across a large number of hives.

## 1 Introduction

Pollinators are one of the most fundamental parts of crop production worldwide (Klein et al., 2007). Without honey bee pollinators, there would be a substantial decrease in both the diversity and yield of our crops, which includes most common produce (van der Sluijs & Vaage, 2016). As a model organism, bees are also often studied through controlled behavioral experiments, as they exhibit complex responses to many environmental factors, many of which are yet to be fully understood. A colony of bees coordinate its efforts to maintain the overall health, with different types of bees tasked for various purposes. One of the signature modality of characterizing bee behavior is through the buzzing frequencies emitted through the vibration of the wings, which can correlate with various properties of the surroundings, including temperature, potentially allowing for a descriptive 'image' of the hive in terms of strength (Howard et al., 2013; Ruttner, 1988).

However, despite what is known about honey bees behavior and their importance in agriculture and natural diversity, there remains a substantial gap between controlled academic studies and the field practices carried out (López-Uribe & Simone-Finstrom, 2019). In particular, beekeepers use their long-tenured experience to derive heuristics for maintaining colonies, which necessitates frequent visual inspections of each frame of every box, many of which making up a single hive. During each inspection, beekeepers visually examine each frame and note any deformities, changes in colony size, amount of stored food, and amount of brood maintained by the bees. This process is labor intensive, limiting the number of hives that can be managed effectively. As growing risk factors make human inspection more difficult at scale, computational methods are needed in tracking changing hive dynamics on a faster timescale and allowing for scalable management. With modern sensing hardware that can record data for months and scalable modeling with state-of-the-art tools in machine learning, we can potentially start tackling some of challenges facing the management of our pollinators, a key player in ensuring food security for the future.

## 2 BACKGROUND AND RELATED WORKS

Our work falls broadly in applied machine learning within computational ethology, where automated data collection methods and machine learning models are developed to monitor and characterize biological species in natural or controlled settings (Anderson & Perona, 2014). In the context of honey bees, while there has been substantial work characterizing bee behavior through controlled audio, image, and video data collection with classical signal processing methods, there has not been a large-scale effort studying how current techniques in deep learning can be applied at scale to the remote-monitoring of beehives in the field.

Part of the challenge lies in data collection. Visual-sensing within beehives is nearly impossible given the current design of boxes used to house bees. These boxes are heavily confined with narrow spaces between many stacked frames for bees to hatch, rear brood, and store food. This makes it difficult to position cameras to capture complete data, without a redesign of existing boxes. Environment sensors, however, can capture information localized to a larger region, such as temperature and humidity. Sound, likewise, can travel across many stacked boxes, which are typically made from wood and have good acoustics. Previous works have explored the possibility of characterizing colony status with audio in highly stereotyped events, such as extremely diseased vs healthy beehives (Robles-Guerrero et al., 2017) or swarming (Krzywoszyja et al., 2018; Ramsey et al., 2020), where the old Queen leaves with a large portion of the original colony. However, we have not seen work that attempt to characterize more sensitive measurements, such as population of beehives, based on audio. We were inspired by these works and the latest advances in hardware sensing and deep learning audio models to collect audio data in a longitudinal setting across many months for a large number of managed hives, and attempt to characterize some of the standard hive inspection items through machine learning.

While audio makes it possible to capture a more complete picture of the inside of a hive, there are still challenges related to data semantics in the context of annotations. Image and video data can be readily processed and labeled post-collection if the objects of interest are recognizable. However, with honey bees, the sound properties captured by microphones are extremely difficult to discriminate, even to experts, due to the fact that the sound is not semantically meaningful, and microphone sensitivity deviations across sensors makes it difficult to compare data across different hives. Thus, it is not possible to retrospectively assign labels to data, making humans inspections during data collection the only source of annotations. As beekeepers cannot inspect a hive frequently due to the large number of hives managed and the potential disturbance caused to the hive, the task becomes few-shot learning.

In low-show learning for audio, various works have highlighted the usefulness of using semi-supervised or unsupervised objectives and/or learning an embedding of audio data, mostly for the purpose of sound classification or speech recognition (Jansen et al., 2020; Lu et al., 2019). These models typically capture semantic differences between different sound sources. We were inspired by the audio classification work with semi-supervised or contrastive-learning objectives to build an architecture that could model our audio and learn an embedding without relying only on task-specific supervision. Unlike previous audio datasets used in prior works, longitudinal data is unlikely to discretize into distinct groups due to the slower continuously shifting dynamics across time on the course of weeks. Therefore, we make the assumption that unlike current audio datasets which contain audio from distinct classes that can be clustered into sub-types, our data more likely occupy a smooth latent space, due to the slow progression in time of changing properties, such as the transition between healthy and low-severity disease, or changes in the size of the population, as bee colonies increase by only around one frame per week during periods of colony growth (Russell et al., 2013; Sakagami & Fukuda, 1968).

## 3 METHODS

**Hive Setup**   Each hive composes of multiple 10-frame standard Langstroth boxes stacked on top of one another, with the internal sensor located in the center frame of the bottom-most box, and the external sensor on the outside side wall of the box. This sensor placement is based on prior knowledge that bees tend to collect near the bottom box first prior to moving up the tower (Winston, 1987). Due to difficulties in obtaining data that would span the spectrum of different colony sizes

without intervention in a timely manner, we set up hives of varying sizes in the beginning to capture a range of populations. This allowed our dataset to span a number of frame sizes, from 1 to 23 for bee frames, and 0 to 11 for brood frames. Aside from these manipulations, all other observed effects, such as progression of disease states, are of natural causes free from human intervention.

## 3.1 DATA COLLECTION

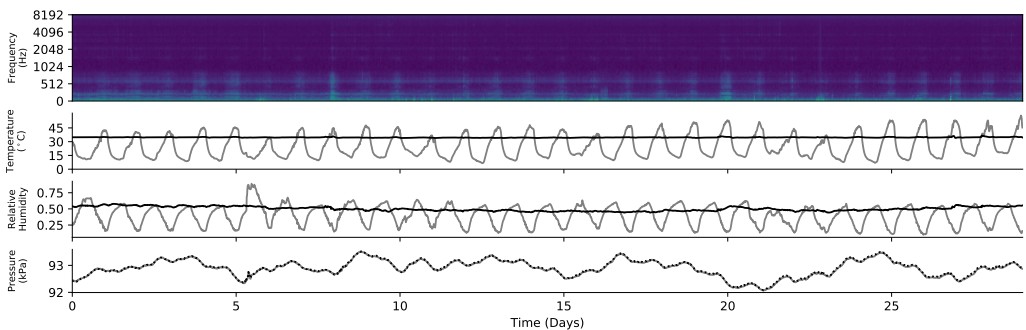

Figure 1: Collected multi-modal data from our custom hardware sensor for one hive across one month. The sensor records a minute-long audio sample and point estimates of the temperature, humidity, and air pressure for both inside and outside the hive every 15 minutes during all hours of day. Black line indicates measurements recorded internal of bee box; gray line indicates external of box. Internal measurements are consistent for temperature and humidity as bees thermoregulate internal hive conditions (Stabentheiner et al., 2010).

**Sensor Data**    Given prior works that showed the possibility of characterizing honey bee colonies through audio, we developed battery-powered sensorbars that can be fitted to a single frame of a standard Langstroth bee box. Each sensor is designed for longitudinal data collection over the span of many weeks on a single charge. The sensor records sub-sampled data every 15 minutes, at all hours of day. Each multi-modal data sample composes of a one-minute audio sample and point estimates of the temperature, humidity, and pressure, for both inside and outside the box (Fig. 1). For the purpose of the daily-snapshot model described in this work, we use data from all days with 96 samples collected. In sum, we have collected ~1000 total days of data, across 26 hives, with up to 180 corresponding human inspection entries. These inspection entries captured information related to hive status, which for our purpose are frames of bees, frames of brood, disease status, and disease severity.

**Inspections**    We used data from one inspector for all collected data used in this work in order to increase annotation consistency. The inspector performed observation in each hive roughly once per week, during which they visually examine each individual frames in all boxes for that hive. The hives are placed 2 meters apart from one another such that cross contamination of audio is unlikely, given the sensor is isolated to within each stack of boxes. For frame labels, the inspector visually examine each frame to determine if that frame is at least 60% covered, given which it would be added to the total frame count. We prevent overcrowding on each frame by introducing empty frames whenever necessary, such that each frame is covered at most up to 90%, as is common practice. This allows us to obtain a lowerbound on the error range of our inspections at around ±20%. During the same inspection, the inspector also check for the presence of any diseases and its severity. Severity scores between none, low, moderate, and severe, where low corresponds to a single observation of diseased bees, moderate for several observations of disease, and severe for prevalent signs of disease.

## 4 GENERATIVE-PREDICTION NETWORK

Given the difficulty of collecting ground truths due to the nature of our data, we sought out to develop a semi-supervised model and leverage our large number of audio samples. Additionally, behavior from bees leaving and returning to beehives means that data from one full-day circadian cycle must be used for predictions in order to model same-day variations. Therefore, we developed a model

trained on hierarchical objectives to allow for modeling both low-level audio features on a minute-long basis, as well any complex temporal dynamics within a given day. We do not consider longer time horizon for this work as the focus is in modeling a snapshot of the hive's current state, not where it will be in the future. Given prior works characterizing beehive sound profiles in lab settings, we know that local audio features are critical, as audio strength along certain known frequencies correlate with different behaviors and types of bees, which could potentially allow for discerning population sizes and disease statuses.

## 4.1 ARCHITECTURE

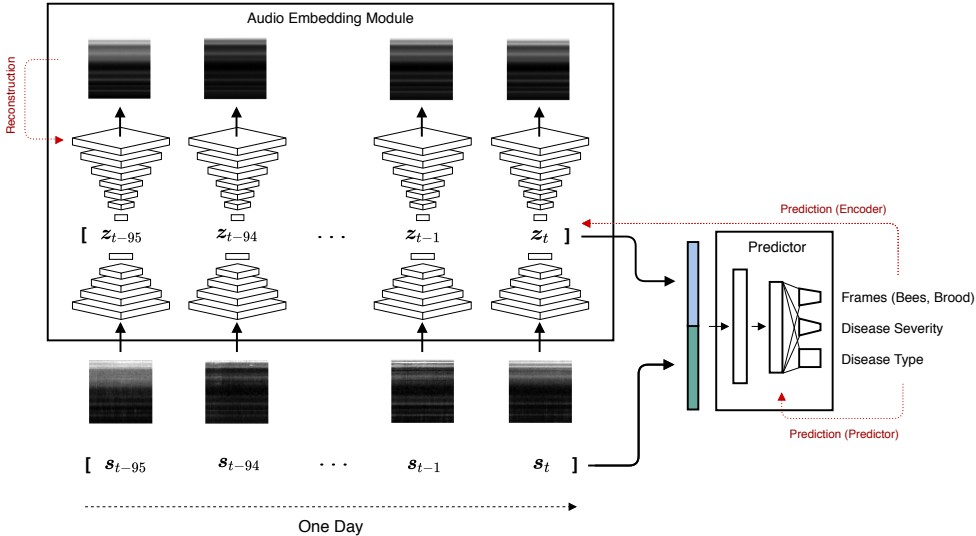

Figure 2: Hierarchical Generative-Prediction Network. $s$: point estimates of environmental factors (temperature, humidity, air pressure), $z$: latent variables. The encoder has 4 convolutional layers with max-pooling in between, and the decoder has 7 transposed-convolutional layers. The Predictor receives concatenated inputs from the encoder's latent variables as well as sensor data for 96 samples across one day. The predictor attaches to multiple heads, sharing parameters to encourage shared-representation learning and regularize based on a multi-task objective.

The model composes of two components in hierarchy and purpose: an **audio embedding module**, and a temporal **prediction module** (Fig. 2). The **embedding module** learns a very low-dimensional representation of each one-minute long audio samples, while sharing its parameters across all 96 samples across each day. Each encoder-decoder is a convolutional variational autoencoder. This embedding module outputs a concatenated audio latent space, which is $96 \times d_z$, representing all samples from the beehive across one day. The embedding module is trained via variational inference on maximizing the log likelihood of the reconstruction, which is a 56 x 56 downsampled mel spectrogram, same as the input. The embedding module is pre-trained to optimize each sample separately, and not capture temporal dynamics explicitly. It is used to learn feature filters that are less dependent on the prediction loss downstream, which can bias the model due to limited data that has assigned inspection entries.

The **predictor** is a shallow feed-forward network, designed to prevent overfitting and model simple daily temporal dynamics. It is trained only on data with matching inspection labels. The predictor takes in all concatenated latent variables from 96 audio samples of each day, along with corresponding 96 samples of environmental data, which includes temperature, humidity, and pressure for inside and outside the box. The sensor input is less important for predicting the momentary population and disease status than to normalize the observed audio, given that activity is known to vary with respect to temperature and humidity. The predictor is then connected to multiple heads, jointly-trained on multi-task objectives. The parameter sharing used in the predictor is also designed to reduce overfitting and to capture similar representations, as the predicted tasks are likely similar in nature (disease and population).

**Objectives**   The embedding module is trained jointly on audio sample reconstruction via the evidence lowerbound (Eq. 1) as well as a global prediction loss across a given day, backpropagated through the latent variables. The generator is pre-trained for $\sim$8000 iterations in order to learn a stable latent representations before prediction gradients are propagated. The predictor is trained via a multi-task prediction losses. This training then continues until all losses have converged and stabilized. The multi-task objective is composed of Huber loss (Eq. 2) for frames and disease severity regressions and categorical cross-entropy for disease classification.

$$\log p(x) \geq \mathcal{L}(x) = \mathbb{E}_{z \sim q(z|x)} \log p(x|z) - D_{\mathrm{KL}}[q(z|x)||p(z)] \tag{1}$$

$$L(y, f(x)) = \left\{ \begin{array}{ll} \frac{1}{2}\left[y - f(x)\right]^2 & \text{for } |y - f(x)| \leq \delta, \\ \delta\left(|y - f(x)| - \delta/2\right) & \text{otherwise.} \end{array} \right. \tag{2}$$

## 4.2   EVALUATION

Due to the nature of our annotation collection, deviations can be expected in our ground truths for both diseases and frames. In particular, inspections often occur on days with incomplete data samples. In order to reduce uncertainty around our inspection entries, we curated our training and validation sets by excluding inspections that were recorded more than two days away from the nearest day with a complete set of sensor data, due to hardware or battery issues causing some days to record incomplete data. This led us to a inspection-paired dataset of 38 samples across 26 hives, spanning 79 days. Recognizing the limited sample size, we carry out 10-fold validation with all models evaluated in our model comparisons. In addition, to reduce the possibility of cross contamination between the training and test set due to sensor similarities, we do not train and validate on the same sensor / hive, even if the datapoints are months apart. This is done despite our sensors not being cross-calibrated, as we wanted to see whether our predictions are able to fully generalize across completely different hives without the need for fine-tuned sensors, which can be costly to implement.

To account for the variance around ground truths of frames, we compute cumulative density functions for percentage differences between all frame predictions and inspections, in order to examine the fraction of predictions that fall within our ground truth error range's lowerbound, which is $\sim \pm 20\%$ of the assigned label (Fig. 4). We compute validation scores for each partitioned validation set, averaged across all 10 groups and for each training iteration, in order to gather a more complete understanding of how each model's training landscape evolves, and also to assess how easily each model overfits (Fig. 9). For all evaluation results, we show mean losses / scores across 10 separate runs, each composed of training a model 10 times on a 10-fold validation scheme.

## 5   RESULTS

### 5.1   MODEL COMPARISONS

We compared several versions of the model with ablation in order to understand the effect of each module or objective on prediction performance (Table 1). First, we compared GPN without the sound-embedding module, and trained it on environment features and hand-crafted audio features, which is the norm in prior literature. These handcrafted audio features include fast Fourier transformed (fft) audio and mean audio amplitude, where fft is sampled across 15 bins between 0 to 2667 Hz, which is just above what prior literature have shown bee related frequencies to be under (Howard et al., 2013; Qandour et al., 2014). We also trained a version of GPN purely on audio without any environment data, which we expect is needed to remove the confounding effects of external environment on bee behavior.

**MLP trained on sensor data and fft features.**   We found that the fully supervised MLP trained on handcrafted features worked well, given a carefully selected bin size, performing slightly worse than GPN trained on spectrograms (Fig. 3). We think this is evidence that the GPN is learning filters across the frequency domain rather than the temporal domain, as frequencies likely matter more than temporal structure within each minute-long sample given the sound lack any fast temporal variations.

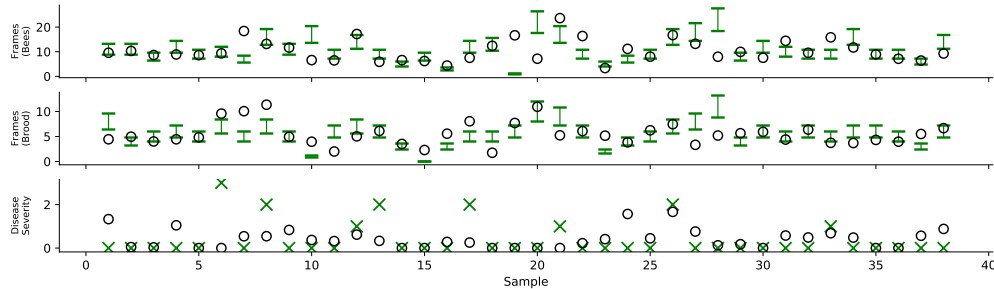

Figure 3: Model predictions on combined 10-fold validation set, where each fold validates on 1-3 hives that the model was not trained on. Ranges / crosses indicate inspection, circles indicates predictions. We indicate frame inspection label with a range of $\pm 20\%$, which represents an approximate lowerbound on the error range of the ground truth. These samples comes from 26 hives sampled across the span of 79 days.

Table 1: Task prediction performance. *Env*: environment data. *fft*: fast Fourier transformed features. *labeled*: data with assigned inspections. *Unlabeled*: all data samples. Metric for frames and disease severity: Huber loss; disease type: accuracy.

| Task | MLP *Env, fft* | GPN *Labeled* | GPN *Unlabeled* |
|------|------|------|------|
| Frames | $0.0206 \pm 2.7e{-}3$ | $0.227 \pm 4.2e{-}1$ | $\mathbf{0.0166} \pm 2.1e{-}3$ |
| Disease Type | $0.757 \pm 1.6e{-}2$ | $0.689 \pm 1e{-}1$ | $\mathbf{0.781} \pm 1.2e{-}2$ |
| Disease Severity | $0.0343 \pm 6.7e{-}3$ | $0.259 \pm 5.1e{-}1$ | $\mathbf{0.0331} \pm 4.7e{-}3$ |

**GPN trained on all vs only labeled data.** As we collect 96 audio samples per day across many months, many of these datapoints are not associated with matching annotations for task predictions. However, given that our model can be trained purely on a generative objective, we compared models training on all of our data on days with complete 96 samples, vs only on the data with assigned inspection entries. By including unlabeled data, we are able to increase our valid dataset from 38 to 856. We found that this model performed better for all tasks, likely attributable to learning more generalizable kernels in the encoders (Fig. 3).

## 5.2 Effects of Environment Modalities on Performance

As this is the first work we have seen using combined non-visual multi-modal data for predicting beehive populations, we wanted to understand which modalities are most important for prediction performance. Thus, we tested the effects of sensor modality exclusion on performance. We trained multiple GPNs, removing each environmental modality from the sensor data used for predictor training, and observe the increases in validation loss or decrease in accuracy (Table 2). We found that when compared to a GPN trained on all sensor modalities in Table 1, removing temperature had the greatest effect on performance, while humidity and pressure had less effect. This is possibly because thermoregulation is an important behavioral trait that is highly correlated with a healthy hive. Since air pressure is unlikely to capture this property, humidity and temperature are likely the most important, with temperature being sufficient. We do believe that these modalities are important for prediction if hives are placed in radically different geographic locations with varying altitudes and seasonal effects, which is beyond the scope of our dataset.

## 5.3 Learned Embedding and Audio Profiles

Due to the nature of the purely observational data resulting in limited samples of diseased states, we examined the generative model's embeddings and outputs from the decoder to evaluate the generative model's ability to model diseased states. In particular, we wanted to understand if it has learned to model variations in audio spectrograms, and if a purely unsupervised model trained on

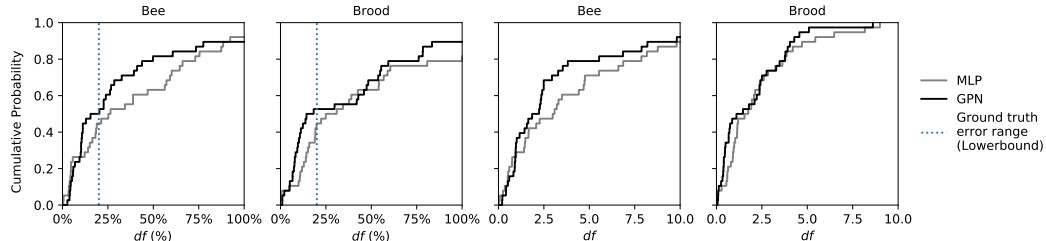

Figure 4: Frame prediction performance comparing supervised MLP on fft audio features and semi-supervised GPN. CDFs of absolute differences and percentage difference shown.

Table 2: Environment modality exclusion's effect on performance. Mean shown with standard deviation computed for 10 runs of each group, validated 10-fold.

| Task | Humidity | Temperature | Pressure |
|---|---|---|---|
| Frames | $0.0189 \pm 1.6e{-}3$ | $0.143 \pm 4.3e{-}1$ | $0.0207 \pm 4e{-}3$ |
| Disease Type | $0.763 \pm 4e{-}9$ | $0.724 \pm 1.4e{-}1$ | $0.763 \pm 1e{-}2$ |
| Disease Severity | $0.0366 \pm 9e{-}2$ | $0.220 \pm 6.7e{-}1$ | $0.0315 \pm 4.9e{-}3$ |

spectrograms alone or with prediction signal would show inherent differences between diseased and healthy states.

We sampled audio spectrograms from the test set and examined how well our embedding model has learned to reconstruct the spectrograms, and in general, we found that the embedding model has learned to model the frequency bands we see in our data (Fig. 5) as well as the magnitude of each band. We also sampled from our latent variables and examined the differences in learned sound profiles extracted from the decoder, and found that the two dimensions correspond to differences across the frequency spectrum (Fig. 6 a).

We also projected audio data from the dataset into the embedding space across each day, representing 96 audio samples $\times$ 2 latent dimensions. For visualization, we then compute PCAs of each full-day latent representation. The first two PCA components captured significant portion of the variance of the embedding space, with PC-1 and PC-2 representing respective 74.17% and 10.61% of the variance. We color-mapped this PCA space with disease severity: healthy, low disease, medium disease, and high disease, with the goal of observing how these diseased states are organized in the latent space. We found that within each disease class there was relatively consistent organization. The audio samples that are classified as healthy occupied a more distinct region within this space, and the samples classified as diseased separated out primarily along PC-1. The low disease severity class was the most discriminated within the space, followed by the medium disease severity, then high disease severity. We were interested in what features of the frequency spectra were learned within the embedding model, thus extracted spectrograms across 24 hour window for each projected point and computed the mean across each disease severity class. We compared these frequency spectra to the corresponding latent variables. Within the healthy audio samples, we observed changes in magnitude and broadening of spectral peaks across the day, which could be associated

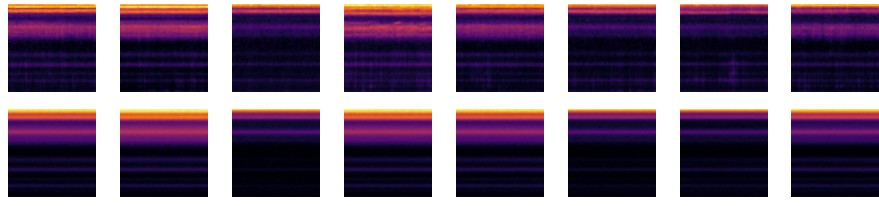

Figure 5: Sampled test spectrograms (top) and reconstructions (bottom).

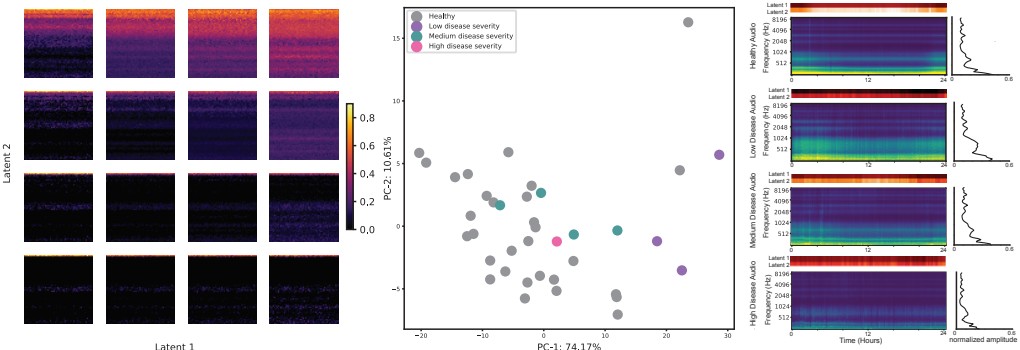

Figure 6: **left**: Sampled latent space decoding. The jointly trained generative model on audio data allows us sample the space occupied by the two latent variables. Distinct audio profiles show the model captured a range of beehive sound profiles. **center**: Visualization of audio samples in latent space across one day reduced to two PCA dimensions, color-mapped to disease severity. **right**: The average frequency spectra for each disease class synchronized across a 1-hour window.

with honey bee circadian rhythms, and similar changes are observed in the corresponding latent variables (Eban-Rothschild & Bloch, 2008; Ramsey et al., 2018). The low disease severity class, which shows the most separation in PCA space, has the most distinct spectrogram signatures. In particular, within the low-disease severity we see the peak at 645 Hz broadening and increasing in magnitude that was consistent across 24 hours. As the disease class progressed from low to medium and high severity, we observed reduction in magnitude and any circadian changes become less apparent. These characteristics appear to be consistent with studies comparing healthy and disease auditory and behavioral modifications within bee hives, and the embedding model is capturing some of the differences in acoustic profiles between the disease severity classes (Kazlauskas et al., 2016; Robles-Guerrero et al., 2017).

## 6 CONCLUSION

We have shown for the first time the potential of using custom deep learning audio models for predicting strengths of beehives based on multi-modal data composed of audio spectrograms and environment sensing data, despite collecting data across 26 sensors in 26 hives without cross-sensor calibration and ground truth uncertainties for downstream prediction tasks. We believe that properly calibrated microphones with improved signal, better sensor placement, and better managed inspection labels would further improve the predictions, reducing the need for frequent human inspections, a major limitation of current beekeeping practices. We have also shown the usefulness of semi-supervised learning as a framework for modeling beehive audio, of which is easy to collect large amounts of samples but difficult to assign inspections, making annotation collection only possible during data collection. The learned latent space allows us to characterize different sound profiles and how they evolve over time, as shown with the progression in disease severity of pollinators in this work. Future efforts may explore how similar frameworks can be used to study the dynamics of beehives across longer time horizons combined with geographic data, and explore its use in forecasting, which can be used to improve labor and time management of commercial beekeeping.

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

# A APPENDIX

## A.1 DATA VALIDATION

We examine the quality of collected sensor data in several ways. For environmental sensor data, which composes of temperature, humidity, and pressure, we developed a dashboard to examine all datapoints to ensure smoothness across time and check all values to ensure they are within plausible range. For audio data, we looked for features that could characterize mean activity across day. Bees have rigid circadian rhythms and are active during the day. By looking at a simple metric such as audio amplitude, we were able to see that the peak amplitude occurred on average around midnight, when bees are mostly in the hive. This is congruent with established literature, which also suggests that bees often vibrate their wings to modulate the hive temperature, which we have verified through the inside temperature sensed for hives with bees compared to without bees (Weiss et al., 2017; Southwick, 1983; Stabentheiner et al., 2010). Boxes with bees have consistent inside temperatures due to thermoregulation by healthy colonies, whereas an empty box we tested captures the same internal and external temperatures.

We also computed Pearson correlation coefficients between pairwise features to verify the plausibility of the linear relationships between our data features (Fig. 8), and trained linear models to predict each feature based on other features. We additionally examined conditional probabilities in order to verify each relationships in greater detail. In summary, we find that there is some sensitivity differences across microphone, other sensor modalities were most likely self and cross-consistent. We also inspect audio spectrograms and remove data samples with failed audio captures.

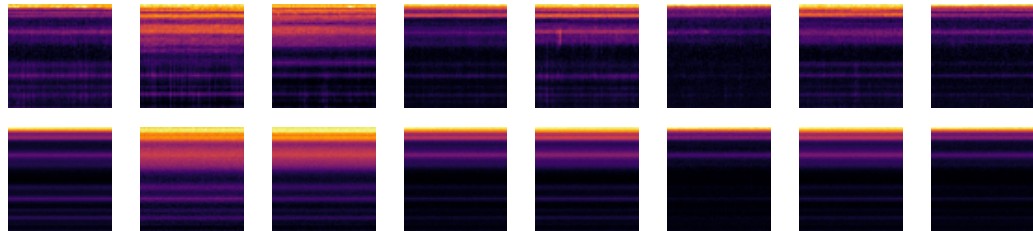

Figure 7: Sampled training spectrograms (top) and reconstructions (bottom).

## A.2 DATA PREPROCESSING

Each 56-second audio sample is converted into a wav file and preprocessed with the python package `Librosa` to obtain a full sized mel spectrogram which is $128 \times 1680$ with a maximum frequency set at 8192 Hz, half of the sampling rate of the data at 16,384. This spectrogram is then downsampled through mean pooling to 61 by 56, with 61 representing the frequency dimension, and 56 representing second-long time bins. Given the spectrogram captures frequencies well beyond bee sounds established in prior literature (Ramsey et al., 2020; Howard et al., 2013; Ferrari et al., 2008), we crop this spectrogram at 56 dimensions, representing a 56 by 56 downsampled spectrogram, which is used to feed into the embedding module after normalizing to between 0 and 1. We did not use some common transformations such as Mel-frequency cepstrum (MFCC), as it enforces speech-dominant priors that do not apply to our data and would likely result in bias or data loss. We also normalize each environmental sensor feature separately and as well as prediction ground truths to between 0 and 1 for more stable training.

## A.3 DATASET PREPARATION

It can be difficult to assign data to each inspection given often inspections tend to correspond to gaps in data collection. We worked with our inspector to understand the nature of each inspection entry for each hive. For each inspection entry, we match that entry to the closest date we have a full 96 samples for. We realized after doing this that a number of inspections could not be matched to any sensor data, and unfortunately had to discard those labels from the dataset. In order to be stringent

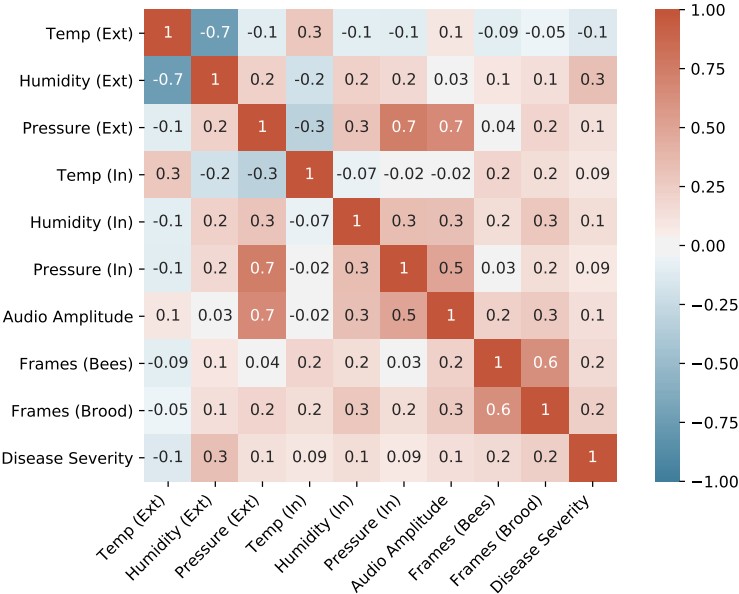

Figure 8: Feature's Pearson correlation coefficients. **Ext**: external sensor, **In**: internal sensor. Aside from several expected strong correlations, there does not appear to be other bivariate linear confounders.

about our label assignment, we only pair labels to data if the difference in time is under 2 days. This led to a dataset of 40 days with labels.

## A.4 HYPERPARAMETERS

**Training**   Adam was used as the optimizer for all objectives. We found that training with 4 objectives was relatively stable, given sufficient pretraining. We used a learning rate of 3e-5 for all except for disease classification, which we found required a slightly larger learning rate to converge at 1e-4. The multiple objectives seems to have regularized the model from overfitting as evident in the validation curves, with the exception of diseases likely because there is significant class imbalance and insufficient number of examples for diseased, due to the nature of the dataset. The number of pretrain iterations was determined empirically based on the decrease in validation loss. This pretraining is useful in order to prevent the network from overfiting to the prediction loss from the very beginning and not learn to model the audio spectrogram.

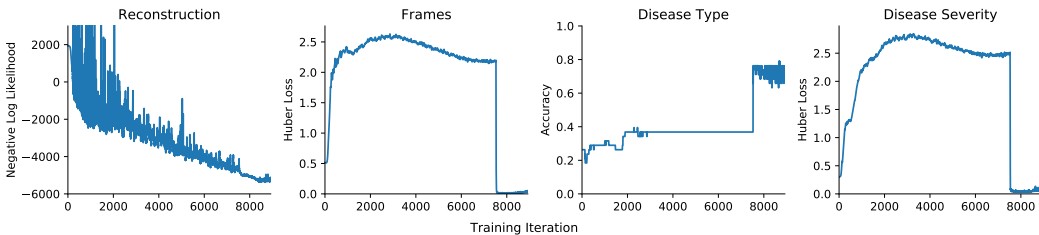

Figure 9: Tracking validation metrics for GPN across training iterations, averaged across 10-folds. Generative model pretraining occupies the bulk of the training iterations, after which the validation loss quickly plateaus.

**Architecture**   We found that the predictor could overfit to the training set very easily, thus decided on using a single layer which attaches to multiple heads to allow for parameter sharing. We swept over different numbers of latent variables for the generative module, and found two latent

variables worked best when considering prediction performance and also reconstruction quality and interpretability of the latent space.

### A.5 AUDIO FEATURES CAPTURED BY THE EMBEDDING MODEL

**Magnitude** After projecting audio data from the dataset into the embedding space across each day, and computing PCAs of each full-day latent representation, we found the first two PCA components captured significant portion of the variance of the embedding space, with PC-1 and PC-2 representing respective 74.17% and 10.61% of the variance. To understand what specific audio features were captured by the embedding model, we swept in ascending order across PC-1 and created a 24 hour average frequency spectra for each sample. We observed the broadening of a peak around 675 Hz and increase in magnitude across all spectra. We plotted integrated magnitudes for frequency spectra from 0 to 8196 Hz against order on PC-1, and found high correlation of magnitude within the defined region to position on PC-1, demonstrating that the embedding model captured variation in magnitude across this frequency range.

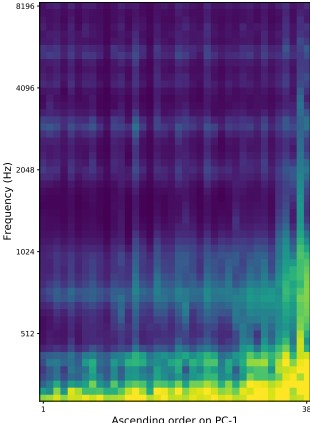 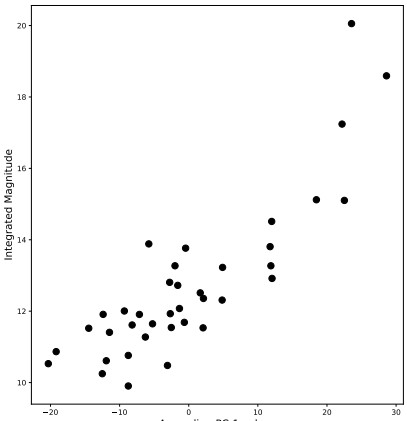

Figure 10: The average 24 frequency spectra for each of the 38 samples within the embedding space organized by ascending value on PC-1. Integrating the magnitude from 0 - 8196 Hz for each point and plotting against the order on PC-1

