# OpenReview forum: "Semi-Supervised Audio Representation Learning for Modeling Beehive Strengths"
_ICLR.cc/2021/Conference — Reject_

### Official Review · AnonReviewer2 · 2020-10-29
**Okish paper. Not an appropriate venue.**

**Rating:** 4
**Confidence:** 5

**Review:**

The paper proposes audio-driven multi-sensory modeling for predicting beehive strengths. The presentation is very clear; it describes what the task is, the data sources are, its limitations etc. And the proposed model based on GPN makes sense. Although, it seems the GPN may not outperform a baseline MLP from Table 1.

Overall, this paper is a demonstration of utilizing machine learning methods and applying them in a plugin-and-test fashion to problems in natural science. Keeping the nature of the venue in mind; my main concern is the relevance and appropriateness of the technical content of this paper to ICLR -- i.e., the authors chose a wrong venue.

---

### Official Review · AnonReviewer4 · 2020-10-29
**off topic paper**

**Rating:** 3
**Confidence:** 4

**Review:**

The paper presents a semi-supervised model to predict the vitality of beehives. The inputs of the model are data from sensors (audio on one hand and environmental on the other hand such as temperature, humidity ...).  The objective is to predict simultaneously 3 values of interest: the frames state of beehives, the potential diseases and their severity. The architecure is composed of two modules, the first one is an auto-encoder in charge of embedding the audio spectrogram in a low latent dimensional space and the second one a MLP to predict the outputs from the latent spectrogram and the environmental data. The paper presents results of the proposed architecture on a small dataset, an ablation study to show the benefits of the auto-encoder module and the role of the environmental data and a latent space analysis to understand the ability of the model to capture relevant audio information linked to the diseases.

In my opinion, the paper is off topic. The presented model is very simple with nothing innovative (except for the application to beehives), the ML aspect of the paper is quite small (just 1 reference is ML oriented, all other references are from biology field) and the choice of the model is not really explained (why not something sequential ?), the experimental data are very small and it is hard to judge the relevance of the approach (only ablation studies is presented, no other approaches/baselines are considerd), I do not see how the approach can be generalized to other similar applications (with multi-modal data, audio combined to environmental time series). I am sure that the subject is interesting and important but I don't think that ICLR is the right conference for this paper.

In addition, the experimental part is hard to follow : the outputs of the model and the human labeling information are not clearly stated (what are the different diseases type ?  it is the same with diseases status ? the diseases severity seems to be discrete - bottom of page 3 - but a continuous Huber loss is used ?). The results are also very quickly analysed : Fig. 3 gives the impression that disease severity is always under estimated when the severity is high, but no analysis is provided. In summary, I can not state if the problem that the authors addres is a hard one or with simpler models same results can be achieved.

Minor remarks : add color to figures 1,4 (and not black vs gray) for more visibility.

---

### Official Review · AnonReviewer5 · 2020-11-05
**Possibly decent contribution in a vertical domain. Uncertain whether it is strong enough for accept**

**Rating:** 5
**Confidence:** 3

**Review:**

Authors have implemented semi-supervised model for inferring beehive characteristics (frames, disease severity, and disease type) based on audio sensor and environmental sensor data.

Contributions:
1. Sensor and environmental data collection from beehives together with sparse inspection data. This may be important contribution to the domain, but not a direct contribution to ICLR.
2. Creating and training semi-supervised model to infer beehive characteristics based on low quantity of data in general and sparse inspection data. In particular, authors propose to use embedding module that is trained for reconstruction. The approach may be applicable to other domains with low quantity of audio data.

I am not an expert in beehive monitoring, so it is hard for me to assess the novelty of the work done as specific to this domain.
Looking from general point of view, the approach seems reasonable while taking into account low amount of data available and even lower amount of inspection data. Although I understand why only low amounts of data are available, the lack of data still raises questions whether the model is accurate enough and useful without larger amounts of data. And if more data were available, should the model architecture change significantly to achieve better accuracy?

Specific questions and concerns:
1. Is there a reason why hive placement is discussed in "inspections" section? I think that placement is independent of inspections.
2. I found the figures to be illustrative, but not very informative. For example, Figure 3 does not tell much except that some predictions are close to actuals and some are quite off. However, this is not really informative. Since the paper is handling a vertical domain, in my opinion, there should be discussion of precision vs. recall and possibly the cost of having false positives vs. false negatives. If you have false positive for disease, the cost would be additional human inspection, while if you have false negative, the cost might be hive loss. I acknowledge that there might be difficulties in addressing this since authors have very few inspection observations.
3. Related to point 2: in Table 1, performance for disease severity and frames are shown by providing Huber loss. It is not clear how that relates into real world metrics that someone monitoring hives would consider. How does Huber loss translate into accuracy or accounting for false positives and false negatives?
4. Figure 4 is not referenced and not discussed in the text.
5. Figure 5 is also illustrative for eyeballing the similarities/differences, but it would be nice to have a quantitative evaluation of similarity possibly across frequencies.
6. Figure 6 right hand side: again it is nice for eyeballing. It seems that latent 2 output is the biggest differentiator between healthy and unhealthy hives. Perhaps this should be investigated and mentioned. Audio outputs seem to be less useful. I would be concerned a bit that eyeballing may mislead that problem can be solved by just looking at the picture and that picture contains enough info for classification.
7. What happens to prediction if audio is excluded and only environmental sensors are used for prediction? Is the accuracy very bad?

In summary, this is a nice proof-of-concept work in a specific vertical domain. It may have some lessons for other domains, but I am not sure it is strong enough for accept.

---

### Author Response · Authors · 2020-11-12
**General response regarding relevance and novelty**

We wanted to thank all reviewers for reading our paper and their comments, especially related to presentation. We acknowledge it’s difficult to assess works at the boundary of two very different fields, and it was also very difficult when we made the decision to submit this to ICLR.

In responding to relevance: we agree that our applied work may not fit in with established areas previously published at ICLR. However, that’s the common case for most works exploring new areas of application. We were aiming to show the possibility of applying semi-supervised training in a new domain, under the conference track "applications in audio, speech, robotics, neuroscience, computational biology, or any other field", as directly quoted from ICLR’s paper call. Unlike many of ML’s existing popular application areas (e.g. standard EEG or medical datasets), applying deep learning to beehives is not well established, as there’s not a set of methods, benchmarks or even standard way to collect data. However, this area is growing rapidly, and will become increasingly common as an application area, leading to more standardized methods. On that end, we believe it’s important to start tackling the problem somewhere, which is why it may seem that there’s less related works cited or methods compared against in this area when you compare against other applied works in well-established domains.

We do wish to address some of the presentation concerns some of you have raised, and also agree that some of the standard measurements of performance were hindered by the size of the dataset. On the other hand, we still think there’s novelty in the cross-modal prediction problem, where labels are collected from a different modality as the sensing itself.

---

### Decision · Program_Chairs · 2021-01-07
**Final Decision**

**Decision:**

Reject

**Comment:**

All Reviewers and myself believe that ICLR may not be the right venue for this paper. Hence, my recommendation is to REJECT it. As a brief summary, I highlight below some pros and cons that arose during the review and meta-review processes.

Pros:
- Important domain, but out of scope of ICLR.
- Collection of sensor and environmental data, which may be potentially hard to collect.

Cons:
- Not the appropriate venue.
- Lack of machine learning novelty.
- Potential lack of generalization of the proposed approach.
- Experimental part is hard to follow.
- Not very informative figures.